# Learning deep abdominal CT registration through adaptive loss weighting and synthetic data generation

**Javier Pérez de Frutos** [1]*, **André Pedersen**[1,2,3], **Egidijus Pelanis**[4], **David Bouget**[1], **Shanmugapriya Survarachakan**[5], **Thomas Langø**[1,6], **Ole-Jakob Elle**[4], **Frank Lindseth**[5]

1 Department of Health Research, SINTEF, Trondheim, Norway, 2 Department of Clinical and Molecular Medicine, Norwegian University of Technology (NTNU), Trondheim, Norway, 3 Clinic of Surgery, St. Olavs hospital, Trondheim University Hospital, Trondheim, Norway, 4 The Intervention Centre, Oslo University Hospital, Oslo, Norway, 5 Department of Computer Science, Norwegian University of Science and Technology (NTNU), Trondheim, Norway, 6 Research Department, Future Operating Room, St. Olavs hospital, Trondheim University Hospital, Trondheim, Norway

* javier.perezdefrutos@sintef.no

## Abstract

### Purpose

This study aims to explore training strategies to improve convolutional neural network-based image-to-image deformable registration for abdominal imaging.

### Methods

Different training strategies, loss functions, and transfer learning schemes were considered. Furthermore, an augmentation layer which generates artificial training image pairs on-the-fly was proposed, in addition to a loss layer that enables dynamic loss weighting.

### Results

Guiding registration using segmentations in the training step proved beneficial for deep-learning-based image registration. Finetuning the pretrained model from the brain MRI data-set to the abdominal CT dataset further improved performance on the latter application, removing the need for a large dataset to yield satisfactory performance. Dynamic loss weighting also marginally improved performance, all without impacting inference runtime.

### Conclusion

Using simple concepts, we improved the performance of a commonly used deep image registration architecture, VoxelMorph. In future work, our framework, DDMR, should be validated on different datasets to further assess its value.

**Data Availability Statement:** The IXI dataset is made available under the Creative Commons CC BY-SA 3.0 license, from https://brain-development.org/ixi-dataset/. The Oslo-CoMet dataset is not

publicly available as per agreement with the data holders. Access can be requested to The Intervention Centre, Oslo University Hospital, Oslo, Norway (contact via Dr. Ole-Jakob Elle, oelle@ous-hf.no), for researchers who meet the criteria for access to confidential data.

**Funding:** This study was supported by the H2020-MSCA-ITN Project No. 722068 HiPerNav; Norwegian National Advisory Unit for Ultrasound and Image-Guided Therapy (St. Olavs hospital, NTNU, SINTEF); SINTEF; St. Olavs hospital; and the Norwegian University of Science and Technology (NTNU). The funders had no role in study design, data collection and analysis, decision to publish, or preparation of the manuscript.

**Competing interests:** The authors have declared that no competing interests exist.

## Introduction

For liver surgery, minimally invasive techniques such as laparoscopy have become as relevant as open surgery [1]. Among other benefits, laparoscopy has shown to yield higher quality of life, shorten recovery time, lessen patient trauma, and reduce blood loss with comparable long-term oncological outcomes [1]. Overcoming challenges from limited field of view to manoeuvrability, and a small work space are the foundations of laparoscopy success. Image-guided navigation platforms aim to ease the burden off the surgeon, by bringing better visualisation techniques to the operating room [2, 3]. Image-to-patient and image-to-image registration techniques (hereafter image registration) are at the core of such platforms to provide clinically valuable visualisation tools. The concept of image registration refers to the alignment of at least two images, matching the location of corresponding features across images in order to express them into a common space. Both rigid and non-rigid registration are the two main strategies to define the alignment between the images. Rigid registration uses affine transformations, which are quicker to compute but less accurate as these are applied globally. Non-rigid registration, also known as deformable registration, defines a diffeomorphism, i.e., a point-to-point correspondence, between the images. However, non-rigid registration comes at the expense of higher computational needs and thus hardware constraints might hinder the development and deployment of such algorithms. In medicine, image registration is mandatory for fusing clinically relevant information across images; groundwork for enabling image-guided navigation during laparoscopic interventions [4, 5]. Additionally, laparoscopic preoperative surgical planning benefits from abdominal computed tomography (CT) to magnetic resonance imaging (MRI) registration to better identify risk areas in a patient's anatomy [6].

During laparoscopic liver surgeries, intraoperative imaging (e.g., video and ultrasound) is routinely used to assist the surgeon in navigating the liver while identifying the location of landmarks. In parenchyma-sparing liver resection (i.e., wedge resection) for colorectal liver metastasis, a minimal safety margin around the lesions is defined to ensure no recurrence and spare healthy tissue [7]. When dealing with narrow margins and close proximity to critical structures, a high accuracy in the registration method employed is paramount to ensure the best patient outcome. Patient-specific cross-modality registration between images of different nature (e.g., CT to MRI) is practised [8], yet being a more complex process compared to mono-modal registration.

The alignment of images can be evaluated through different metrics based either on intensity information from the voxels, shape information from segmentation masks, or spatial information from landmarks' location or relative distances. The most common intensity-based similarity metrics are the normalised cross-correlation (NCC), structural similarity index measure (SSIM), or related variations [9, 10]. For segmentation-based metrics, the most notorious are the Dice similarity coefficient (DSC) and Hausdorff distance (HD) [11]. However, target registration error (TRE) is the gold standard metric for practitioners, conferring a quantitative error measure based on the target lesion location across images [12].

Research on the use of convolutional neural networks (CNNs) for image registration has gained momentum in recent years, motivated by the improvements in hardware and software. One early application of deep learning-based image registration (hereafter deep image registration) was performed by Wu *et al.* [13]. They proposed a network built with two convolutional layers, coupled with principal component analysis as a dimensionality reduction step, to align brain MR scans. Expanding upon the concept, Jaderberg *et al.* [14] introduced the spatial transformer network, including a sampling step for data interpolation, allowing for gradients to be backpropagated. Hence, further enabling neural network deformable image-to-image registration applications. Publications on CNNs for image-registration show a preference for

encoder-decoder architectures like U-Net [15], followed by a spatial transformer network, as can be seen in Quicksilver [16], VoxelMorph [9], and other studies [17]. Mok *et al.* [18] proposed a Laplacian pyramid network for multi-resolution-based MRI registration, enforcing the non-rigid transformation to be diffeomorphic.

The development of weakly-supervised training strategies [19, 20] enabled model training by combining intensity information with other data types (e.g., segmentation masks). Intensity-based unsupervised training for non-rigid registration was explored for abdominal and lung CT [21, 22]. Building cross-modality image registration models through reinforcement learning has also been explored [23]. However, semi-supervised training of convolutional encoder-decoder architectures has been favoured for training registration models and producing the displacement map [24].

In our study, the focus is brought towards improving the training scheme of deep neural networks for deformable image registration to cater more easily to use-cases with limited data. We narrowed the scope to mono-modal registration, and the investigation of transfer learning across image modalities and anatomies. Our proposed main contributions are:

- an augmentation layer for on-the-fly data augmentation (compatible with TensorFlow GPU computational graphs), which includes generation of ground truth samples for non-rigid image registration, based on thin plate splines (TPS), removing the need for pre-computation and storage of augmented copies on disk,

- an uncertainty weighting loss layer to enable adaptive multi-task learning in a weakly-supervised learning approach,

- and the validation of a cross-anatomy and cross-modality transfer learning approach for image registration with scarce data.

## Materials and methods

### Dataset

In this study, two datasets were selected for conducting the experiments: the Information eXtraction from Images (IXI) dataset and Laparoscopic Versus Open Resection for Colorectal Liver Metastases: The Oslo-CoMet Randomized Controlled Trial dataset [1, 25].

The IXI dataset contains 578 T1-weighted head MR scans from healthy subjects collected from three different hospitals in London. This dataset is made available under the Creative Commons CC BY-SA 3.0 license. Only T1-weighted MRIs were used in this study, but other MRI sequences such as T2 and proton density are also available. Using the advanced normalization tools (ANTs) [26], the T1 images were registered to the symmetric Montreal Neurological Institute ICBM2009a atlas, to subsequently obtain the segmentation masks of 29 different regions of the brain. Ultimately, left and right parcels were merged together resulting in a collection of 17 labels (see the online resource Table A in S1 Appendix). The data was then stratified into three cohorts: training (n = 407), validation (n = 87), and test (n = 88) sets.

The Oslo-CoMet trial dataset, compiled by the Intervention Centre, Oslo University Hospital (Norway), contains 60 contrast-enhanced CTs. The trial protocol for this study was approved by the Regional Ethical Committee of South Eastern Norway (REK Sør-Øst B 2011/1285) and the Data Protection Officer of Oslo University Hospital (Clinicaltrials.org identifier NCT01516710). Informed written consent was obtained from all participants included in the study. Manual delineations of the liver parenchyma, i.e., liver segmentation masks, were available as part of the Oslo-CoMet dataset [4]. Additionally, an approximate segmentation of the vascular structures was obtained using the segmentation model available in the public

livermask tool [27]. The data was then stratified into three cohorts: training (n = 41), validation (n = 8), and test (n = 11) sets.

## Preprocessing

Before the training phase, both CT and MR images, as well as the segmentation masks, were resampled to an isotropic resolution of 1 mm$^3$ and resized to $128 \times 128 \times 128$ voxels. Additionally, the CT images were cropped around the liver mask before the resampling step. Cubic spline interpolation was used for resampling the intensity images, whereas segmentations were interpolated using nearest neighbour. The segmentation masks were stored as categorical 8-bit unsigned integer single-channel images, to enable rapid batch generation during training.

To overcome the scarcity of image registration datasets for algorithm development, we propose an augmentation layer, implemented in TensorFlow [28], to generate artificial moving images during training. The augmentation layer allows for data augmentation and preprocessing. The layer features gamma (0.5 to 2) and brightness augmentation (±20%), rotation, and rigid and non-rigid transformations, to generate the moving images. Data preprocessing includes resizing and intensity normalisation to the range [0, 1]. The maximum displacements, rigid and non-rigid, can be constrained to mimic real-case scenarios. In our case, 30 mm and 6 mm respectively. Rotation was limited to 10°, for any of the three coordinate axes.

The non-rigid deformation was achieved using TPS applied on an $8 \times 8 \times 8$ grid, with a configurable maximum displacement. Rigid transformations include rotation and translation.

## Model architecture

The baseline architecture consists of a modified VoxelMorph model [9], based on a U-Net [29] variant. The model was used to predict the displacement map, as depicted in Fig 1. After the augmentation step, the fixed ($I_f$) and the generated moving ($I_m$) images were concatenated into a two-channel volumetric image and fed to the VoxelMorph model. The model returns the displacement map ($\Phi$) i.e., a volume image with three channels, which describes the relative displacement of each voxel along each of the three coordinate axes. Finally, the predicted fixed image ($I_p$) is reconstructed by interpolating voxels on the moving image at the locations defined by the displacement map. This way, the model can be trained by comparing the predicted image with the original fixed image.

When provided, the segmentations ($S_m$) are likewise updated using the same displacement map. The symmetric U-Net architecture was designed with six contraction blocks featuring 32, 64, 128, 256, 512, and 1024 convolution filters respectively. Each contracting block consisted of a convolution with kernel size $3 \times 3 \times 3$ and a LeakyReLU activation function, followed by max pooling with stride 2. The decoder blocks consisted of a convolution and a LeakyReLU activation function, followed by a nearest neighbour interpolation upsampling layer. The output convolutional layer, named Head in Fig 1, was set to two consecutive convolutions of 16 filters with LeakyReLU activation function. A convolution layer with three filters was used as the output layer. This produces a displacement map with the same size as the input images and three channels, one for each displacement dimension.

## Model training

The registration model was trained in a weakly-supervised manner, as proposed by Hu *et al.* [19]. Instead of evaluating the displacement map directly as in traditional supervised training, only the final registration results were assessed during training.

Due to the complexity of the task at hand, a single loss function would provide limited insight of the registration result, therefore a combination of well-known loss functions was

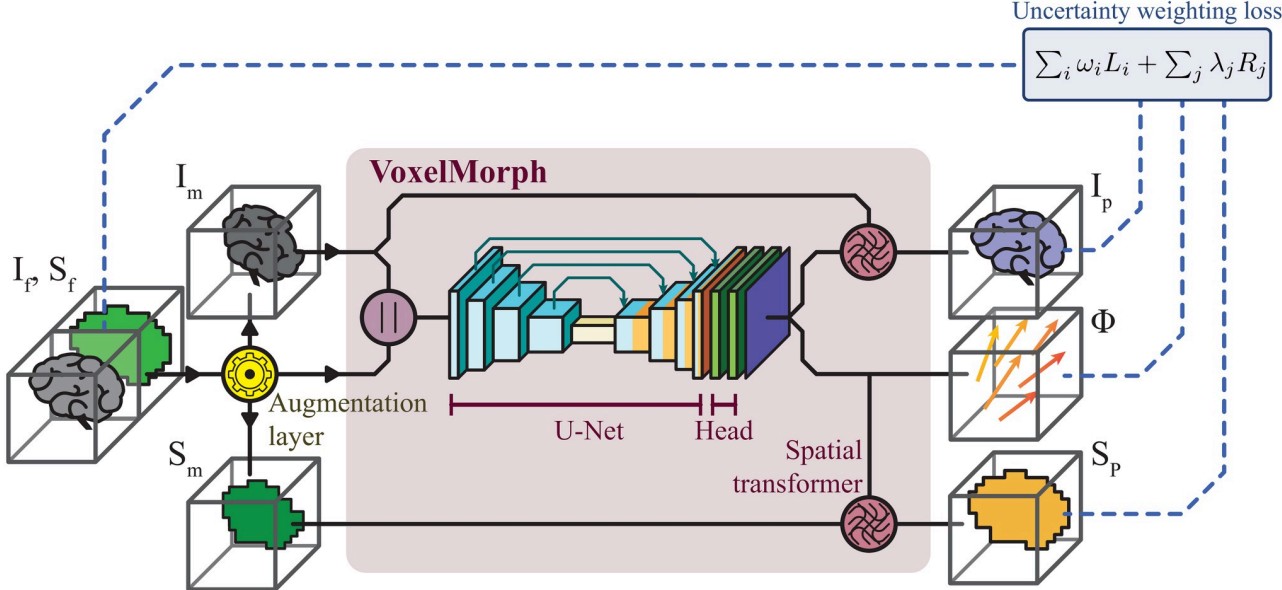

**Fig 1. Proposed deep image registration pipeline.** Generation of artificial moving images on-the-fly, prediction of the displacement map using a modified U-Net, and finding the optimal loss weighting automatically using uncertainty weighting.

deemed necessary. Balancing the contribution of these operators can be challenging, time consuming, and prone to errors. We therefore used uncertainty weighting (UW) [30], which combines losses as a weighted sum and enables the loss weights to be tuned dynamically during backpropagation. Our loss function $\mathcal{L}$ was implemented as a custom layer, and consists of a weighted sum of $N$ loss functions $L$ and $M$ regularisers $\mathcal{R}$:

$$\mathcal{L}(\mathbf{y}_t, \mathbf{y}_p) = \sum_{i=1}^{N} \omega_i L_i(\mathbf{y}_t, \mathbf{y}_p) + \sum_{j=1}^{M} \lambda_j \mathcal{R}_j \tag{1}$$

such that $\Sigma\omega_i = \Sigma\lambda_i = 1$. By default, the weights $\omega_i$ and $\lambda_i$ were initialised to equally contribute in the weighted sum, but can be set manually from a priori knowledge on initial loss and regularisation values. In our experiments, the default initialisation for the loss weights was used, except for the regularisation term, which was initialised to $5 \times 10^{-3}$.

For training the neural networks we used the Adam optimiser. Gradient accumulation was performed to overcome memory constraints and enable larger batch training. The batch size was set to one, but artificially increased by accumulating eight mini-batch gradients. The learning rate was set to $10^{-3}$ with a scheduler to decrease by 10 whenever the validation loss plateaued with a patience of 10. The models were trained using a custom training pipeline. Training curves can be found in Figs A-D in S3 Appendix. The training was limited to $10^5$ epochs, and manually stopped if the model stopped converging. The model with the lowest validation loss was saved.

## Experiments

The experiments were conducted on an Ubuntu 18.04 Linux desktop computer with an Intel®Xeon®Silver 4110 CPU with 16 cores, 64 GB of RAM, an NVIDIA Quadro P5000 (16 GB VRAM) dedicated GPU, and SSD hard-drive. Our framework, DDMR, used to conduct the experiments was implemented in Python 3.6 using TensorFlow v1.14. To accelerate the

**Table 1. Configurations trained on both the IXI and Oslo-CoMet datasets.**

| Design | Model | Loss function |
|---:|---|---|
| **BL-N** | Baseline | NCC |
| **BL-NS** | Baseline | NCC, SSIM |
| **SG-ND** | Segmentation-guided | NCC, DSC |
| **SG-NSD** | Segmentation-guided | NCC, SSIM, DSC |
| **UW-NSD** | Uncertainty weighting | NCC, SSIM, DSC |
| **UW-NSDH** | Uncertainty weighting | NCC, SSIM, DSC, HD |

BL: baseline, SG: segmentation-guided, UW: uncertainty weighting, N: normalized cross correlation, S: structural similarity index measure, D: Dice similarity coefficient, H: Hausdorff distance.

research within the field, the source code is made openly available on GitHub (https://github.com/jpdefrutos/DDMR).

As aforesaid, our aim was to improve the training phase of image registration for CNN models. To that extent, four experiments were carried out:

**(i) Ablation study** Different training strategies and loss function combinations were evaluated to identify the key components in deep image registration. Three different training strategies were considered, all using weakly-supervised learning: 1) the baseline (BL) using only intensity information, 2) adding segmentation guidance (SG) to the baseline, and 3) adding uncertainty weighting (UW) to the segmentation-guided approach. For all experiments, the input size and CNN backbone were tuned prior and kept fixed. All designs are described in Table 1 and evaluated on both the IXI and Oslo-CoMet datasets. Six loss weighting schemes were tested, using different combinations of loss functions, including both intensity and segmentation-based loss functions. For the second experiment, the entire model was finetuned directly or in two steps, i.e., by first finetuning the decoder, keeping the encoder frozen, and then finetuning the full model. A learning rate of $10^{-4}$ was used when performing transfer learning.

**(ii) Transfer learning** To assess the benefit finetuning for deep image registration to applications with a small number of samples available, e.g., abdominal CT registration.

**(iii) Baseline comparison** The trained models were evaluated against a traditional registration framework (ANTs), to better understand the potential of deep image registration. This experiment was performed only on the Oslo-CoMet dataset, as ANTs was used to generate the segmentations on the IXI dataset. Two different configuration were tested: symmetric normalisation (SyN), with mutual information as optimisation metric, and SyN with cross-correlation as metric (SyNCC).

**(iv) Training runtime** The last experiment was conducted to assess the impact of the augmentation layer (see Fig A in S2 Appendix). The GPU resources were monitored during training. Only the second epoch was considered, as the first one served as warm-up.

## Evaluation metrics

The evaluation was done on the test sets of the IXI and Oslo-CoMet datasets, for which the fixed-moving image pairs were generated in advance, such that the same image pairs were used across methods during evaluation. After inference, the displacement maps were resampled back to isotropic resolution using piecewise 3D linear interpolation. The final predictions

were then evaluated using four sets of metrics to cover all angles. Image similarity was assessed under computation of NCC and SSIM metrics. Segmentations were converted into one-hot encoding and evaluated using DSC, HD, and HD95 (95th percentile of HD) measured in millimetres. The background class was excluded in the segmentation metrics computation. For image registration, TRE was estimated using the centroids of the segmentation masks of the fixed image and the predicted image, also measured in millimetres. In addition, the methods were compared in terms of inference runtime, only measuring the prediction and application of the displacement map, as all other operations were the same between the methods.

Five sets of statistical tests were conducted to further assess: 1) performance contrasts between designs, 2) benefit of transfer learning, 3) benefit of segmentation-guiding, 4) benefit of uncertainty weighting, and 5) performance contrasts between the baseline and segmentation-guided models, and the traditional methods in ANTs (SyN and SyNCC). For the tests, the TRE metric was used, as it is considered the gold standard for surgical practitioners. Test 1) was conducted on the evaluations of the IXI test set, whereas tests 2) to 5) were performed using the results on the Oslo-CoMet test dataset only. Furthermore, for the tests only involving the Oslo-CoMet dataset, the two-step transfer learning approach was used as reference, as these models showed the best results.

For the statistical test 1), multiple pairwise Tukey's range tests were conducted on the IXI experiment comparing all the designs described in Table 1. For 2), benefit of transfer learning was assessed using a one-sided, non-parametric test (Mann-Whitney U test), comparing the differences between the BL-N, SG-NSD, and UW-NSD designs on the Oslo-CoMet experience. The three generated p-values were corrected for multiple comparison using the Benjamini-Hochberg method. For 3)-5), Mann-Whitney U tests were conducted comparing BL-N and SG-NSD to assess benefit of segmentation-guiding, SG-NSD and UW-NSD to assess benefit for uncertainty weighting, and BL-N and SG-NSD against SyN and SyNCC to assess the difference in performance between deep image registration and traditional image registration solutions, respectively. The two p-values were corrected using the Benjamini-Hochberg method. The results for all five sets of tests can be found in S4 Appendix.

The Python libraries statsmodels (v0.12.2) [31] and SciPy (v1.5.4) [32] were used for the statistic computations. A significance level of 0.05 was used to determine statistical significance.

## Results

In Tables 2 to 5, the best performing methods in terms of individual performance metrics, i.e., most optimal mean and lowest standard deviation, were highlighted in bold. See the online resources for additional tables and figures not presented in this manuscript.

On the IXI dataset, fusing NCC and SSIM improved performance in terms of intensity-based metrics for the baseline model, whereas segmentation metrics were degraded (see Table 2). Adding segmentation-guiding drastically increased performance across all metrics compared to the baseline. Minor improvement was observed using uncertainty weighting, whereas adding the Hausdorff loss was not beneficial. In terms of TRE, multiple pairwise Tukey's range tests confirmed the benefit of segmentation-guiding ($p < 0.001$), however, no significant improvement was observed in introducing uncertainty-weighing ($p = 0.9$). The complete pairwise comparison can be found in Table A in S1 Appendix.

On the Oslo-CoMet dataset, a similar trend as for the IXI dataset was observed (see Table 3). However, in this case, the baseline model was more competitive, especially in terms of intensity-based metrics. Nonetheless, segmentation-guiding was still better overall ($p < 0.001$ in terms of TRE), as well as uncertainty weighting ($p = 0.0093$ in terms of TRE).

**Table 2. Evaluation of the models trained on the IXI dataset.**

| Model | SSIM | NCC | DSC | HD | HD95 | TRE | Runtime |
|---|---|---|---|---|---|---|---|
| BL-N | 0.23±0.16 | 0.52±0.12 | 0.03±0.01 | 109.71±26.19 | 100.26±27.91 | 29.47±8.46 | 0.83±0.77 |
| BL-NS | 0.25±0.16 | **0.53±0.12** | 0.02±0.01 | 145.36±22.41 | 138.19±23.48 | 30.06±9.07 | 0.73±0.56 |
| SG-ND | 0.45±0.24 | 0.46±0.10 | 0.61±0.08 | 4.64±1.37 | 2.15±0.54 | 1.08±0.39 | 0.82±0.58 |
| SG-NSD | 0.46±0.24 | 0.46±0.11 | 0.61±0.07 | 4.54±1.42 | 2.10±0.49 | 1.07±0.37 | 0.74±0.64 |
| UW-NSD | **0.47±0.24** | 0.46±0.11 | **0.63±0.08** | **4.44±1.40** | **2.03±0.51** | **0.97±0.36** | **0.72±0.59** |
| UW-NSDH | **0.47±0.24** | 0.46±0.11 | 0.61±0.07 | 4.63±1.49 | 2.14±0.52 | 1.06±0.36 | 0.75±0.59 |
| Unregistered | 0.45±0.21 | 0.24±0.07 | 0.07±0.06 | 21.77±5.15 | 18.73±4.88 | 11.53±3.01 | - |

The best performing methods for each metric are highlighted in bold.

Finetuning the entire model trained on the IXI dataset to the Oslo-CoMet dataset (see Table 4 transfer nonfrozen), yielded similar intensity-based metrics overall, but drastically improved the segmentation-guided and uncertainty weighted models in terms of segmentation metrics. The best performing models overall used uncertainty weighting. When finetuning the model in two steps, the uncertainty weighted designs were further improved to some extent (see Table 5 frozen encoder). The statistical analysis 2) shows significance improvement of the TRE when performing transfer learning and finetuning in two steps, for the UW-NSD model ($p = 0.0014$) and SG-NSD ($p = 0.0021$) models. No statistical significance was observed for the BL-N ($p = 0.8608$).

The traditional methods, SyN and SyNCC, performed well on the Oslo-CoMet test set. However, the segmentation masks were distorted, in particular the vascular segmentations mask (see Fig C in S5 Appendix). Both methods performed similarly, but the SyNCC was considerably slower. Segmentation guidance was deemed critical in obtaining better performance in terms of TRE ($p < 0.001$) compared to SyN and SyNCC. Yet no significant difference was observed on the baseline models ($p = 0.5845$). All deep learning models had similar inference runtimes of less than one second, which was expected as the final inference model architectures were identical. On average, the CNN-based methods were $\sim 13\times$ and $\sim 421\times$ faster than SyN and SyNCC, respectively. The deep learning models struggled with image reconstruction, unlike ANTs (see Fig C in S5 Appendix). For instance, anatomical structures outside the segmentation masks were poorly reconstructed in the predicted image, e.g., the spine of the patient.

**Table 3. Evaluation of the models trained on the Oslo-CoMet dataset.**

| Model | SSIM | NCC | DSC | HD | HD95 | TRE | Runtime |
|---|---|---|---|---|---|---|---|
| BL-N | 0.52±0.10 | **0.20±0.07** | 0.23±0.09 | 54.10±7.22 | 30.36±3.58 | 18.11±7.62 | 0.78±1.50 |
| BL-NS | **0.62±0.13** | 0.17±0.07 | 0.29±0.07 | 37.69±8.04 | 22.06±5.17 | 13.95±4.78 | **0.76±1.44** |
| SG-ND | 0.55±0.15 | 0.16±0.06 | 0.38±0.14 | **22.03±8.27** | **12.74±6.12** | **7.60±3.96** | 0.76±1.46 |
| SG-NSD | 0.58±0.13 | 0.12±0.07 | **0.35±0.07** | 25.22±7.92 | 14.49±4.22 | 8.91±3.08 | 0.77±1.49 |
| UW-NSD | 0.54±0.13 | 0.11±0.06 | 0.26±0.07 | 25.08±6.67 | 18.47±5.34 | 11.52±3.32 | 0.77±1.49 |
| UW-NSDH | 0.59±0.14 | 0.14±0.06 | 0.35±0.11 | 24.49±8.67 | 14.57±5.93 | 8.34±4.31 | 0.78±1.50 |
| Unregistered | 0.60±0.13 | 0.09±0.05 | 0.24±0.08 | 24.60±5.56 | 19.06±4.89 | 11.86±2.75 | - |

The best performing methods for each metric are highlighted in bold.

**Table 4. Evaluation of models trained on the Oslo-CoMet dataset from finetuning the entire architecture.**

| Model | SSIM | NCC | DSC | HD | HD95 | TRE | Runtime |
|---|---|---|---|---|---|---|---|
| BL-N | 0.52±0.08 | **0.17±0.07** | 0.23±0.07 | 57.98±5.36 | 33.00±5.14 | 24.09±5.92 | 0.77±1.45 |
| BL-NS | **0.61±0.09** | 0.16±0.07 | 0.14±0.03 | 82.91±6.96 | 59.94±6.41 | 34.41±13.03 | 0.77±1.46 |
| SG-ND | 0.56±0.13 | 0.14±0.07 | 0.43±0.09 | 15.81±5.56 | 9.05±3.18 | 5.89±3.10 | 0.79±1.56 |
| SG-NSD | 0.58±0.13 | 0.14±0.07 | 0.42±0.10 | 16.26±6.37 | 9.50±3.51 | 5.84±3.01 | 0.76±1.48 |
| UW-NSD | 0.58±0.12 | 0.14±0.06 | **0.48±0.11** | 15.53±5.80 | **7.84±3.17** | 4.05±2.41 | **0.76±1.47** |
| UW-NSDH | 0.59±0.12 | 0.14±0.06 | 0.47±0.10 | **15.29±5.65** | 7.91±2.82 | **3.95±2.09** | 0.78±1.51 |
| Unregistered | 0.60±0.13 | 0.09±0.05 | 0.24±0.08 | 24.60±5.56 | 19.06±4.89 | 11.86±2.75 | - |

The best performing methods for each metric are highlighted in bold.

The use of the augmentation layer resulted in a negligible increase in training runtime of 7.7% per epoch and 0.47% ($\sim$74 MB of 16 GB) increase in GPU memory usage (see Fig A in S2 Appendix).

## Discussion

Development of CNNs for image registration is challenging, especially when data is scarce. We therefore developed a framework called DDMR to train deep registration models, which we have evaluated through an ablation study. By pretraining a model on a larger dataset, we found that performance can be greatly improved using transfer learning, even if the source domain is from a different image modality or anatomic origin. Through the development of novel augmentation and loss weighting layers, training was simplified by generating artificial moving images on-the-fly, removing the need to store augmented samples on disk, while simultaneously learning to weigh losses in a dynamic fashion. Furthermore, by guiding registration using automatically generated segmentations and adaptive loss weighting, registration performance was enhanced. In addition, negligible increase in inference runtime and GPU memory usage was observed. The added-value of our method lies in the use of generic concepts, which can therefore leverage most deep learning-based registration designs.

From Tables 2 to 5, segmentation guidance boosts the performance of the image registration both on the SG and UW models, further confirmed by the results of the performance contrast analysis shown in Table A in S4 Appendix ($p < 0.001$), and Figs A-C in S5 Appendix,

**Table 5. Evaluation of the models trained on the Oslo-CoMet dataset from finetuning in two steps.**

| Model | SSIM | NCC | DSC | HD | HD95 | TRE | Runtime |
|---|---|---|---|---|---|---|---|
| BL-N | 0.52±0.07 | **0.19±0.07** | 0.24±0.06 | 60.92±26.06 | 39.96±30.25 | 22.34±8.60 | 0.79±1.52 |
| BL-NS | **0.62±0.10** | 0.17±0.07 | 0.14±0.04 | 85.71±6.40 | 60.93±4.15 | 32.84±11.90 | 0.76±1.45 |
| SG-ND | 0.56±0.12 | 0.14±0.07 | 0.44±0.09 | 16.12±5.29 | 8.87±2.94 | 5.12±2.52 | **0.77±1.48** |
| SG-NSD | 0.58±0.12 | 0.15±0.07 | 0.43±0.08 | 16.93±6.50 | 9.17±3.02 | 5.21±2.40 | 0.77±1.49 |
| UW-NSD | 0.60±0.11 | 0.15±0.06 | **0.53±0.13** | 15.13±5.68 | **6.97±2.83** | **3.40±1.91** | **0.77±1.48** |
| UW-NSDH | 0.60±0.12 | 0.15±0.06 | 0.50±0.12 | **14.79±5.79** | 7.37±2.99 | 3.55±2.14 | **0.77±1.48** |
| SyN | 0.61±0.13 | 0.20±0.07 | 0.49±0.01 | 17.93±3.44 | 9.62±1.57 | 22.34±4.96 | 10.01±3.69 |
| SyNCC | 0.63±0.13 | 0.20±0.07 | 0.49±0.01 | 18.59±2.99 | 9.64±1.61 | 22.31±5.04 | 323.81±87.13 |
| Unregistered | 0.60±0.13 | 0.09±0.05 | 0.24±0.08 | 24.60±5.56 | 19.06±4.89 | 11.86±2.75 | - |

The best performing methods for each metric are highlighted in bold.

found in the online resources. The introduction of landmarks to guide the training, in the form of boundaries of the different segmentation masks, allows for a better understanding of the regions occupied by each anatomical structure. This observation is drawn by the improvement of the segmentation-based metrics on the finetuned models (see Table 5 frozen encoder). And further confirmed by the statistical tests 2) and 3), in which the Mann-Whitney U test showed significant difference for the segmentation guidance ($p < 0.001$) and uncertainty weighting models ($p = 0.0093$) (see Table C in S4 Appendix). No statistical difference was observed for the baseline models (see Table 5 frozen encoder). A larger dataset is required to fully assess the significance of the transfer learning, as only eleven test samples were available for this study.

Surprisingly, adding HD to the set of losses had limited effect on the performance. We believe this is due to HD being sensitive to outliers and minor annotation errors, which is likely to happen as the annotations used in this study were automatically generated. Furthermore, NCC proved to be a well-suited intensity-based loss function, with no real benefit of adding an additional intensity-based loss function such as SSIM.

From studying the adaptive loss weights evolution during training (see Figs E-L in S3 Appendix), it is possible to deduce an interpretation regarding influence and benefit from each loss component over the network. Evidently, SSIM was favoured over NCC during training, even though SSIM was deemed less useful for image registration compared to NCC. A rationale can be hypothesised from SSIM being easier to optimise, being a perception-based loss. Interestingly, the loss weight curves all seemed to follow the same pattern. Upweighted losses are linearly increased until a plateau is reached and the opposite behaviour happens for the downweighted losses. This may indicate that uncertainty weighting lacks the capacity of task prioritisation, which could have been helpful at a later stage in training. Such an approach has been proposed in the literature [33], simply not for similar image registration tasks. Hence, a comparison of other multi-task learning designs might be worth investigating in future work.

From Tables 4 and 5 it can be observed the benefit of using segmentation guidance for training deep registration models. Furthermore, when compared to the traditional method ANTs, using SyN and SyNCC, a significant improvement on TRE is observed ($p < 0.001$) (Table D in S4 Appendix), with differences close to 17 mm on the Oslo-CoMet test set. Further improved using uncertainty weighting. No significant value was observed between the baseline model and ANTs ($p = 0.5845$), which shows that naive image-only training is not enough for the model to understand the registration task. Not surprisingly, runtimes of the deep registration models are dramatically better than those of ANTs, taking the latter up to five minutes on average using the SyNCC configuration.

A sizeable downside in training CNNs for image registration remains the long training runtime. Having access to pretrained models in order to perform transfer learning alleviates this issue, but the substantial amount of training data required, and in our use case annotated data, persists as another tremendous drawback.

Once deployed, such registration models often fail to generalise to other anatomies, imaging modalities, and data shifts in general, resulting in ad hoc solutions. As part of future work investigations, developing more generic deep image registration models would be of interest, tackling both training and deployment shortcomings.

In this study, only synthetic moving images and mostly algorithm-based annotations were used for evaluation. To verify the clinical relevance of the proposed models, a dataset with manual delineations of structures both for the fixed and moving images, and with clinically relevant movements, is required. To illustrate this situation, Table E in S4 Appendix shows a comparison between manual and automatic segmentations of the parenchyma and

the vascular structures on the Oslo-CoMet test set images. Both DSC and HD95 are reported. A good concordance between the automatic and manual parenchyma segmentations can be observed. However, vascular segmentation poses a more challenging problem for automatic methods to tackle. In future work, assessment of the impact of vascular segmentations of diverse quality could be considered. This investigation would require the delineation of the entire training set, which itself is extremely challenging and was thus deemed outside the scope of this study. Nevertheless, such investigation is of definite value and should be part of future works, additionally including human qualitative evaluation of the clinical relevance.

The sole focus on mono-modal registration can be considered as a limitation from our work. Especially when selecting the loss functions. For instance, in multi-modal registration it is common to use mutual information. Hence, investigating the translation between mono and multi-modal designs is of value to assess applicability over various registration tasks. The recent introduction of the new Learn2Reg challenge dataset [24] represents an adequate alley for further investigation over this aspect. While the U-Net architecture, used in this study, is not recent, a substantial number of publications have favoured it for image registration, as shown to outperform vision transformers on smaller datasets [34]. Alternatively, generative adversarial models should be tested, as these networks have shown to produce more realistic looking images [35]. Self-attention [36] for encoding anatomical information, or graph-based neural networks [37] for improved vascular segmentation-guided registration, are concepts that also should be considered in future work.

## Conclusion

In the presented study, we demonstrated that registration models can be improved through transfer learning and adaptive loss weighting even with minimal data without manual annotations. The proposed framework DDMR also enables on-the-fly generation of artificial moving images, without the need to store copies on disk. In future work, DDMR should be validated on data of other anatomies and imaging modalities to further assess its benefit.

## Supporting information

**S1 Appendix. Additional data details.** Additional details about the segmentations produced for the IXI dataset.
(PDF)

**S2 Appendix. Resources impact of the augmentation layer.** Details on the GPU resources usage by the proposed augmentation layer.
(PDF)

**S3 Appendix. Training curves.** Figures of the training curves, as well as the adaptive loss weighting.
(PDF)

**S4 Appendix. Statistical analysis.** Results of the statistical tests described in the manuscript, and additional comparison between manually and automatically generated segmentations.
(PDF)

**S5 Appendix. Qualitative results.** Examples of predictions on the IXI and Oslo-CoMet test datasets.
(PDF)

**S1 File. Evaluation metrics.** Collection of the evaluation metrics of the proposed models when tested on both the IXI and the Oslo-CoMet test datasets.
(CSV)

## Author Contributions

**Conceptualization:** Javier Pérez de Frutos.

**Data curation:** Egidijus Pelanis, David Bouget.

**Formal analysis:** Javier Pérez de Frutos, André Pedersen.

**Funding acquisition:** Thomas Langø, Ole-Jakob Elle, Frank Lindseth.

**Investigation:** Javier Pérez de Frutos, André Pedersen.

**Methodology:** Javier Pérez de Frutos, André Pedersen.

**Software:** Javier Pérez de Frutos, André Pedersen, Shanmugapriya Survarachakan.

**Supervision:** Thomas Langø, Ole-Jakob Elle, Frank Lindseth.

**Validation:** Javier Pérez de Frutos, André Pedersen.

**Writing – original draft:** Javier Pérez de Frutos, André Pedersen, Egidijus Pelanis, David Bouget.

**Writing – review & editing:** Shanmugapriya Survarachakan, Thomas Langø, Ole-Jakob Elle, Frank Lindseth.

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
