## [Decision Letter · Decision Letter 0]

18 Jan 2023

PONE-D-22-33384Train smarter, not harder: learning deep abdominal CT registration on scarce dataPLOS ONE

Dear Dr. Pérez de Frutos,

Thank you for submitting your manuscript to PLOS ONE. After careful consideration, we feel that it has merit but does not fully meet PLOS ONE’s publication criteria as it currently stands. Therefore, we invite you to submit a revised version of the manuscript that addresses the points raised during the review process.

We agree with the reviewers' comments about the clarity and general quality of the manuscript.

We also believe we can accept the manuscript after minor modifications in accordance with their comments.

Among others, the authors should modify the title and add some additional statistical tests.

We look forward to receiving your revised manuscript.

Kind regards,

Paolo Cazzaniga

Academic Editor

PLOS ONE

Journal Requirements:

"This study was supported by the H2020-MSCA-ITN Project No. 722068 HiPerNav; Norwegian National Advisory Unit for Ultrasound and Image-Guided Therapy (St. Olavs hospital, NTNU, SINTEF); SINTEF; St. Olavs hospital; and the Norwegian University of Science and Technology (NTNU)."

Reviewers' comments:

Reviewer's Responses to Questions

**Comments to the Author**

1. Is the manuscript technically sound, and do the data support the conclusions?

Reviewer #1: Yes

Reviewer #2: Yes

2. Has the statistical analysis been performed appropriately and rigorously? 

Reviewer #1: N/A

Reviewer #2: Yes

3. Have the authors made all data underlying the findings in their manuscript fully available?

Reviewer #1: Yes

Reviewer #2: Yes

4. Is the manuscript presented in an intelligible fashion and written in standard English?

Reviewer #1: Yes

Reviewer #2: Yes

5. Review Comments to the Author

Reviewer #1: The paper presents an approach to designing and training models for medical image registration. The authors employ multiple concepts and techniques to prepare and validate their experiments. The manuscript is well-written and generally clear in reception. I like the discussion in particular, it sounds fair and reasonable. My remarks are as follows:

1. The paper title is a bit pretentious. The title of likely every research involving ablation study on machine learning training settings and/or hyperparameters could start with "Train smarter, not harder". I suggest focusing on the actual scope of the study, especially since the workflow covers not only abdominal CT data.

2. Congratulations to the Authors on the successful implementation of the augmentation scheme, but the paper does not convince me that this is a significant contribution. The argument (repeated multiple times) about no need to store the data after augmentation on disk is not convincing too. As far as I know, the training schemes in multiple environments offer the same thing by default.

3. The reader may be interested in some methodology details, e.g., how do you interpolate the data in resampling and resizing (lines 90-91)?

4. I'm confused with the description in lines 119-120: convolution filters are a part of convolutional layers, not max-pooling blocks.

5. Please describe N and M in Eq. (1).

6. Please provide units for metrics like HD or TRE. Are these millimeters or pixel/voxel-size-based?

7. Lines 280+: yes, I support the statement on the necessity to collect expert delineations for a reliable evaluation. I even think that with a generally scarce dataset, the Authors could put some effort into annotating a small portion of the data (5-10 cases?) and report the results in the current manuscript. Please consider such an improvement.

8. Change formatting of axis tick labels in Figs. S6-S13. The loss weights look weird in a "4.50e-1" format, it's just "0.45".

Reviewer #2: This paper investigates different methods to improve the registration of medical images using convolutional neural networks. Different training strategies, loss functions, and transfer learning schemes are examined.

The paper focuses on the registration of sparse medical data using deep learning. It is an important step in the development of new machine-learning-based methods in the medical field.

The framework used (based on VoxelMorph), the U-Net and the use of the segmentations during training are explained very clearly and precisely, so that it is also easy to implement. Figure 1 is also very well-structured and thus helps a lot in understanding the framework used.

Some additional comments are listed as follows:

1) You work on deformable image registration. However, the word "deformable" did not appear anywhere in your paper. My advice would be to mention the term "deformable" in the beginning of the paper, so that it is clear which transformation (rigid, affine, deformable,...) you are dealing with - for example see your source [9].

2) Section Results: In tables 2-5, one more column could be added: Values of the metrics before registration - to better examine the potential of deep learning image registration and to check if all models succeeded in improving the values of the unregistered data (image-based and segmentation-based similarity metrics).

3) I would also recommend to do a significance test and to mark the results in tables 2-5.

6. PLOS authors have the option to publish the peer review history of their article (what does this mean?). If published, this will include your full peer review and any attached files.

Reviewer #1: No

Reviewer #2: No

---

## [Author Response · Author response to Decision Letter 0]

3 Feb 2023

A detailed reply to the Editor and reviewers, as well as a description of the changes performed in the manuscript, is available in the attached file 'Response to reviewers.pdf'

---

## [Decision Letter · Decision Letter 1]

8 Feb 2023

Learning deep abdominal CT registration through adaptive loss weighting and synthetic data generation

PONE-D-22-33384R1

Dear Dr. Pérez de Frutos,

We’re pleased to inform you that your manuscript has been judged scientifically suitable for publication and will be formally accepted for publication once it meets all outstanding technical requirements.

Kind regards,

Paolo Cazzaniga

Academic Editor

PLOS ONE

Additional Editor Comments (optional):

All minor comments have been address, thus the manuscript can be accepted in its current form

Reviewers' comments:

Reviewer's Responses to Questions

**Comments to the Author**

1. If the authors have adequately addressed your comments raised in a previous round of review and you feel that this manuscript is now acceptable for publication, you may indicate that here to bypass the “Comments to the Author” section, enter your conflict of interest statement in the “Confidential to Editor” section, and submit your "Accept" recommendation.

Reviewer #1: All comments have been addressed

Reviewer #2: All comments have been addressed

2. Is the manuscript technically sound, and do the data support the conclusions?

Reviewer #1: Yes

Reviewer #2: Yes

3. Has the statistical analysis been performed appropriately and rigorously? 

Reviewer #1: N/A

Reviewer #2: Yes

4. Have the authors made all data underlying the findings in their manuscript fully available?

Reviewer #1: Yes

Reviewer #2: Yes

5. Is the manuscript presented in an intelligible fashion and written in standard English?

Reviewer #1: Yes

Reviewer #2: Yes

6. Review Comments to the Author

Reviewer #1: Dear Authors,

thank you for your responses, good job.

Imprimatur!

Reviewer #2: (No Response)

7. PLOS authors have the option to publish the peer review history of their article (what does this mean?). If published, this will include your full peer review and any attached files.

Reviewer #1: No

Reviewer #2: No

---

## [Editor Report · Acceptance letter]

15 Feb 2023

PONE-D-22-33384R1 

Learning deep abdominal CT registration through adaptive loss weighting and synthetic data generation 

Dear Dr. Pérez de Frutos:

I'm pleased to inform you that your manuscript has been deemed suitable for publication in PLOS ONE. Congratulations! Your manuscript is now with our production department. 

Kind regards, 

on behalf of

Dr. Paolo Cazzaniga 

Academic Editor

PLOS ONE